# Plasmonic Enhanced Nanocrystal Infrared Photodetectors

**DOI:** 10.3390/ma16083216

**Published:** 2023-04-19

**Authors:** Naiquan Yan, Yanyan Qiu, Xubing He, Xin Tang, Qun Hao, Menglu Chen

**Affiliations:** 1School of Optics and Photonics, Beijing Institute of Technology, Beijing 100081, China; 2Yangtze Delta Region Academy of Beijing Institute of Technology, Jiaxing 314019, China; 3School of Optoelectronic Engineering, Changchun University of Science and Technology, Changchun 130022, China

**Keywords:** infrared photodetectors, nanomaterials, plasmonic structures, surface plasmon enhancement

## Abstract

Low-dimensional nanomaterials are widely investigated in infrared photodetectors (PDs) due to their excellent optical and electrical properties. To further improve the PDs property like quantum efficiency, metallic microstructures are commonly used, which could squeeze light into sub-diffraction volumes for enhanced absorption through surface plasma exciton resonance effects. In recent years, plasmonic enhanced nanocrystal infrared PDs have shown excellent performance and attracted much research interest. In this paper, we summarize the progress in plasmonic enhanced nanocrystal infrared PDs based on different metallic structures. We also discuss challenges and prospects in this field.

## 1. Introduction

Infrared wavelength contains thermal information invisible to human eyes. Photodetectors (PDs) could convert optical signals into electrical signals which can be easily measured. Therefore, infrared PDs are widely used in night vision imaging [1], disease diagnosis [2], atmospheric remote sensing [3], vehicle driving [4], etc. The potential applications of infrared PDs are shown in Figure 1. Nanomaterials, such as two-dimensional (2D) nanomaterials (graphene, black phosphorus, MoS_2_, etc.) [5,6,7,8], one-dimensional (1D) nanomaterials (nanowires or nanotubes) [9,10,11,12] and zero-dimensional (0D) nanomaterials (quantum dots) [13,14,15,16,17,18,19,20,21], exhibiting excellent electrical and optical characteristics, are widely investigated for photodetection. Compared with traditional bulk materials, the properties of nanomaterials are easier to control and can be modified by chemical doping, electrical regulation and physical adsorption. Hence, benefiting from the excellent optoelectronic properties and easy fabrication, infrared PDs based on low-dimensional nanomaterials are developing rapidly. Still, additional efforts are needed to achieve higher device performance, such as faster response speed, higher specific detectivity, more efficient photoelectric conversion, and smaller device footprints.

Surface plasmon nanostructures have the feature of breaking the diffraction limit, which provides a new route for the study of high-performance PDs. In 1902, when Wood irradiated a metal grating with polarized light of continuous spectra, an anomalous diffraction phenomenon was observed on the reflection spectrum, namely “Wood Anomalies” [22]. In 1941, Fano’s team found that “Wood Anomalies” were related to electromagnetic wave resonance at the interface of metal and medium [23]. In 1957, Ritchie’s team first proposed the concept of surface plasmon [24]. In recent decades, surface plasmon resonance has attracted researchers’ interest and has gradually been used to enhance the performance of PDs [25,26,27,28,29]. To be specific, surface plasmon refers to an electromagnetic oscillation mode transmitted along the metal surface, which is generated by the interaction between the external electromagnetic field and the free electrons on the metal surface. This effect can not only transmit in the form of surface plasmon, increasing the transmission path of light in the material, but also form local surface plasmon resonance, resulting in a significant near-field enhancement effect. By taking advantage of the unique optical properties of surface plasmon structures, the light absorption and responsivity of nano-optoelectronic devices can be effectively improved.

Plasmonic enhanced PDs can enhance the local electric field intensity, which is conducive to the effective separation of electrons and holes [30,31]. Therefore, the integration of metal nanostructures with nanomaterials-based infrared PDs can enhance the light absorption of semiconductor nanomaterials, resulting in improved external quantum efficiency. The wavelength of surface plasmon resonance can be varied by changing the size, geometry and dielectric environment of metal nanostructures to adjust the absorption wavelength. Besides, due to diverse surface plasmon resonance modes, many different designs of metal nanostructures are provided for the realization of PDs with excellent performance.

In this article, we review the recent research on plasmonic enhanced nanocrystal infrared PDs. According to the difference between plasmon-enhanced structures, we divide the research into four categories: hole array structures, grating structures, nanoparticles and metal array structures. Depending on the dimension of nanomaterials used for detectors, the four kinds of infrared PDs are further discussed separately. The progress in plasmonic enhanced nanocrystal infrared PDs is summarized in Figure 2.

## 2. Types of Infrared Photodetectors

For infrared PDs with the metal micro-nano structure, when the frequency of the incident infrared light matches the resonant frequency of the surface plasma excitons of the metal micro-nano structure, a large number of photons are absorbed, resulting in a surface plasma exciton resonance and the corresponding resonant absorption peak. By altering the parameters of the periodic metal micro-nano structures and the absorbing layer mediums, the magnitude, position and half-peak width of the resonance absorption peak can be changed.

Therefore, the addition of plasmon-enhanced structures can effectively enhance the light absorption of semiconductor nanomaterials layers and improve the detection performance of infrared PDs. In recent decades, there have been many reports about plasmonic enhanced nanocrystal infrared PDs. In this review, we divide the type of infrared PDs based on the plasmon-enhanced structures and nanomaterials.

### 2.1. Infrared Photodetectors with Hole Array Structures

In 1998, Ebbesen et al. first discovered that metal films with subwavelength hole arrays showed pretty unusual light transmission phenomena [32]. Since then, PDs based on subwavelength hole arrays have attracted the attention of researchers. The transmission enhancement mechanism of periodic subwavelength holes is an important part of the field of surface plasmon resonance. The plasma waves are generated by excitation through an array of periodic holes, resulting in a transmission spectrum that correlates with the size of the structure. Under the resonant wavelength condition, transmission enhancement occurs because more energy passes through the holes. Experimental studies have proved that the transmission enhancement effect is related to various parameters of the subwavelength hole array structure, such as the structural period, hole shape, metal film thickness, etc. Different preparation methods and conditions can be employed in experiments to adjust the size and shape of metal nanoholes, which can be applied to detectors operating at different wavelengths.

#### 2.1.1. 2D Materials-Based Infrared Photodetectors with Hole Array Structures

2D materials include graphene, black phosphorus (BP), transition metal dichalcogenides (TMDCs), etc. Thanks to their advantages, such as high carrier mobility and room-temperature operation, 2D material PDs have become an important research direction in recent years [33]. However, such PDs have low light absorptivity due to the ultra-thin thickness of 2D materials. It has been found that the near-field enhancement effect of the surface iso-excited element structure can effectively improve the optical field density. Therefore, combining the surface iso-excited element structure with 2D material-based PDs should be able to significantly improve the device’s performance.

In 2010, Wu et al. [34] used a 2D metal hole array to enhance the performance of quantum well (QWs) detectors. By perforating the Au film with a periodic hole array, the incident infrared light was converted into surface plasma waves, exciting the intersubband transition of carriers in the QWs. With this structure, the responsivity and detectivity of the device were increased to 7 A/W and 7.4 × 10^10^ Jones, respectively (Figure 3a). They also found that the position and intensity of the resonance peak varied with the period of the hole array, while changing the diameter of the hole only affected the intensity of the resonance peak. Except for circular hole arrays, researchers have also designed other shapes of hole arrays. In 2018, Venuthurumilli et al. [35] utilized bowtie aperture antennas to enhance the inherent polarization selectivity of BP (Figure 3b). As a result, a high photocurrent ratio (armchair to zigzag) of 8.7 was achieved.

Using focused ion beam technology to carve out an array of nanopores in an Au film, in 2020, Zhou et al. [36] fabricated a high-efficiency, polarization-specific and wavelength-sensitive optical modulated PD (Figure 3c). This Au thin film with a nanohole array served as both the metal electrode and the light-harvesting antenna. The external quantum efficiency (EQE) of the nano-diodes was ~4%, several orders of magnitude higher than other similar devices. In 2021, Shabbir et al. [37] proposed an ultrasensitive mid-wave infrared (MWIR) PD model consisting of nanopatterned graphene and vanadium dioxide (VO_2_) hybridized heterostructures (Figure 3d). This PD reached an absorption of nearly 100% in monolayer graphene on account of localized surface plasmons around the patterned circular holes. Taking advantage of the phase change of the thin VO_2_ film, the device achieved a high responsivity of ~10^5^ V/W, a detectivity exceeding ~10^10^ Jones, and a sensitivity of noise equivalent power close to room temperature.

#### 2.1.2. 1D Materials-Based Infrared Photodetectors with Hole Array Structures

1D nanomaterials, such as nanowires, nanotubes and nanorods, have gained much attention in the photoelectric conversion field due to the large specific surface area and interesting electrical and optical properties. The excitation of surface plasmon by metal nanostructures can effectively increase the light absorption of 1D materials, thus improving the photocurrent and other performance of PDs based on 1D materials.

In 2011, Senanayake et al. [38] proposed a novel PD based on surface plasmon field enhancements in nanopillar arrays, whose peak responsivity was up to 0.28 A/W (Figure 4a). The responsivity spectrum could be designed by introducing metal hole arrays working as a 2D plasmonic crystal, so that surface plasmon polariton Bloch waves could couple the incident light into the nanopillars. Moreover, the periodicity, shape of nanoholes and metal dielectric function could be designed to absorb light in different spectral regions. Exploiting vertically grown nanopillar photodiodes, in 2016, Lee et al. [39] reported an InAsSb nanowire-array-based PD with a high quantum efficiency of 29% at 2390 nm (Figure 4b). The high quantum efficiency was achieved because of partially coating the nanopillars with metal, which excited localized surface plasmon resonance (LSPR), resulting in a strong absorption. Compared with the bare nanopillar, this PD exhibited a 37-fold absorption enhancement at 2440 nm. Based on a similar structure, in 2018, Ren et al. [40] demonstrated a short-wave infrared (SWIR) InAs nanowire PD with a spectral response range of 1.2–2.5 µm, consisting of vertically oriented selective-area InAs nanowire photoabsorber arrays on the InP substrate (Figure 4c). The dark current density at a reverse bias of 0.5 V was as low as approximately 130 mA/cm^−2^, comparable to the best commercial or research InAs p-i-n photodiodes. However, the peak detectivity of this nanowire PD at SWIR was low, only close to 2.5 × 10^7^ Jones. The surface plasmon resonance could be enhanced by improving the passivation quality and adjusting the geometric structure of the nanowire device, thereby increasing the absorption and detectivity.

In the same year, Ren et al. [41] proposed an InAsSb nanowire-plasmonic PD operating in the MWIR region (Figure 4d). They first developed an ex-situ passivation technique using (NH_4_)_2_S/Al_2_O_3_ to reduce the nonradiative recombination on the InAsSb nanowire surface. Then, they developed a fabrication process based on dry etching to expose the top surface of nanowires whose sidewalls were completely covered by the Al_2_O_3_ shell. The top of the exposed nanowires functioned as a shadow mask that created uniform nanohole arrays on the top of the Au contact. These nanohole arrays excited surface plasmon modes, which strongly coupled the incident light to the top of the nanowires. However, the PD’s detectivity was low. To confirm the feasibility of high detectivity of PDs based on 1D materials, in 2019, Ren et al. [42] fabricated InAs(Sb)-InP nanowire-plasmonic heterojunction photodiodes to obtain a high room-temperature detectivity in SWIR and MWIR regions (Figure 4e). Optical couplings via an efficient three-dimensional (3D) plasmonic excitonic grating compensated for the loss of light absorption, thus breaking the balance between high responsivity and low dark current. What’s more, a peak detectivity of 3.5 × 10^10^ Jones was observed, and the plasmonic exciton peak D^*^ could be tuned by simply varying the exposure height of the nanowires’ top section. This work demonstrated that nanowires-based PDs could perform better than their planar device counterparts in SWIR and MWIR photodetection regions under certain conditions.

#### 2.1.3. 0D Materials-based Infrared Photodetectors with Hole Array Structures

Quantum dots (QDs) are typical representatives of 0D materials. The quantum effect will become more obvious when reducing the nanoparticle sizes, where the nanoparticles would show unique optical and electrical properties. PDs based on QDs have great potential in the photoelectric detection field due to their low cost, adjustable spectral response range, excellent detection performance, etc., [43]. With the development of plasmon structures, the combination with QDs PDs is gradually becoming popular.

In 2010, Chang et al. [44] successfully realized the integration of an Au 2D hole array (2DHA) structure with semiconductor InAs QDs (Figure 5a). With proper arrays’ parameters, the infrared light response and detectivity at the plasma resonance could reach more than 100% enhancement. Moreover, the design of this 2DHA was suitable for large-scale manufacture and did not degrade the noise current characteristics of PDs. In 2013, Liu et al. [45] measured the transmittance of the plasmonic structure on the surface of 2DHA and explored the connections between transmittance and 2DHA structural parameters (Figure 5b). They also found that the photocurrent of the 10 active QD layers infrared photodetectors (10-QDIPs) and 20 active QD layers infrared photodetectors (20-QDIPs) showed different dependence on the hole diameter of the 2DHA structures. When the hole diameter was greater than 1.6 µm, the photocurrent of 10-QDIPs was saturated and started to decrease, while the photocurrent of 20-QDIPs increased linearly with the hole diameter. The appropriate aperture of the 2DHA surface plasmonic structure could achieve more than six times photocurrent enhancement compared with the Ref-QDIP.

In addition to PDs based on the combination of InGaAs materials with 2D nanohole arrays, there are also many surface plasmon-enhanced detectors based on Si-based Ge materials. In 2017, Yakimov et al. [46] combined a periodic 2DHA monolithic perforated in the Au film with Ge/Si QDs-based infrared photodetectors (QDIPs) (Figure 5c), making the QDIP wavelength selective and responsive in the MWIR region. When the normal light was incident, and the period and aperture of the plasma array were 1.6 μm and 1 μm, respectively, the device showed more than 30 times enhanced responsivity at the wavelength of 5.4 μm. In the same year, Yakimov et al. [47] coupled a similar 2D plasma structure with a Ge/SiGe QDIP, which included self-assembled Ge QDs grown on a virtual SiGe substrate (Figure 5d). The influence of the light’s incident direction on the top side and the substrate side on QDIP was consistent with the previous research. Substrate side illumination significantly increased the direct coupling between light and plasma QDIP, leading to a two-fold increase in photocurrent. At zero bias and 90 K, the responsivity of 40 mAW^−1^ and peak detectivity of 1.4 × 10^11^ cmHz^1/2^W^−1^ were determined at 4 μm wavelength. Through comparison and analysis, in 2018, Yakimov et al. [48] found that the conventional subwavelength hole array perforated in the Au film on the top of QDIP could enhance the detector more effectively than the Au disk array (Figure 5e). This was because the in-plane component of the near field vector and the *E_z_* electric field component were the main influence factors of the surface plasmon wave electric field in the 2D metal disk arrays (2DDA) grating and the 2DHA plasma structure, respectively. At the same time, the *z*-polarized electromagnetic radiation contributed to a larger oscillator strength for the bound-to-continuum transitions of holes in Ge/Si QDs. Hence, the PD with 2DHA showed better performance. To improve the photoresponse of Ge/Si QDIP with limited absorption layer thickness, in 2020, Yakimov et al. [49] introduced a hybrid metal-dielectric metasurface with a regular array of Si pillars into the QDIP (Figure 5f). They observed about 4-fold photoresponse enhancement with the hybrid metasurface device relative to a common plasmonic design with a 2D metal hole array. Compared with the bare QDIP, the peak responsivity of the hybrid detector at 4.4 μm wavelength was increased by 15 times.

### 2.2. Infrared Photodetectors with Grating Structures

Apart from common 2D hole array structures, there are also 1D metal grating structures, including linear gratings and circular gratings [50]. The interaction between incident light and metal gratings generates the surface plasmon resonance effect, making the light enter the substrate through the sub-wavelength hole. Metal grating structures can enhance the light transmission of the sub-wavelength hole and improve the ability of the device to absorb the incident light, thus enhancing the performance of PDs.

#### 2.2.1. 2D Materials-Based Infrared Photodetectors with Grating Structures

Based on the deep silver (Ag) grating, in 2014, Zhao et al. [51] proposed a method to enhance graphene absorption (Figure 6a), which could generate strong local electric fields at magnetic resonance or magnetopolaron (MPs), greatly increasing the absorption rate of the single-atom graphene layer to nearly 70%. The next year, the team theoretically studied a hybrid plasma system containing graphene ribbon arrays on periodic metal gratings (Figure 6b). The results showed that the local resonance in the metal grating could be coupled to the plasma resonance in the graphene band, thereby significantly enhancing the absorption in graphene [52]. However, they did not experiment further. In 2018, Zhang et al. [53] presented an absorption-enhancing graphene-based long-wave infrared (LWIR) PD with similar lattice structures (Figure 6c). The device, consisting of metasurface magnetic plasmons and electric plasma resonators, excited SPPs on the graphene surface. The peak absorption of the graphene was confirmed to exceed 67.2% and could be anticipated to reach 83.7% if a lossless dielectric was applied.

In 2018, Li et al. [54] demonstrated a SiO_2_ grating integrated graphene PD with an ultra-thin metal layer (Figure 6d). The interaction between the grating and graphene led to a high optical responsivity of 195.4 mAW^−1^, two orders higher than that of a pure monolayer graphene-based PD. The highlight of the device was that the integrated grating was an ultra-thin layer of metal instead of a block. In 2019, Huang et al. [55] showed a 20-layer BP PD with a nanoscale plasmonic grating structure (Figure 6e), exhibiting high absorption and responsivity. After comparing different grating materials and rationally designing the geometric parameters of the nanoscale grating structure, the light absorption of the proposed BP PD reached 89.8% at the resonance wavelength of 714 nm. Besides, the cut-off wavelength of the device was extended to the MWIR band, and the responsivity was as high as 60.94 A/W. In 2022, Lien et al. [56] integrated an absorption-enhancing resonant metal-insulator-metal (MIM) metasurface grating with a thin-film BP PD (Figure 6f). The MIM structure could significantly improve the BP detector’s responsivity from 12 to 77 mAW^−1^ in the MWIR range. The increased absorption of the device was observed, and the resonance absorption peak could be adjusted by changing the structural parameters of the MIM grating.

#### 2.2.2. 1D Materials-Based Infrared Photodetectors with Grating Structures

The coupling of 1D materials-based PDs with grating structures is also gradually developed. In 2021, Zhang et al. [57] made a dual-band plasma-enhanced Ge PD with alternate arrays of Au gratings and Ge nanowires (Figure 7a). They found that appropriate alternate flask-shaped Au grating-Ge nanowire arrays could enhance the plasmon effect in the infrared band of 1310–1550 nm. As a result, the responsivity of the PD at 1310 and 1550 nm reached 0.75 and 0.62 A/W, respectively, improved by nearly 100% compared with the device without Au grating-Ge nanowire arrays. Such structures were widely used in photovoltaic devices, for example, in the solar cell industry. In 2021, Zhang et al. [58] demonstrated a laterally oriented GaAs p-i-n nanowire solar cell based on the Ag grating structure (Figure 7b). Due to the synergistic effect of the grating diffraction and plasmon polaron excitation, the optical absorption of nanowires was greatly enhanced. At the optimal grating period, the conversion efficiency of the device was increased from 8.7% to 14.7%. Besides, the grating enhanced the absorption of LWIR light and extended the absorption cut-off wavelength of the ultra-thin nanowires, achieving a significant efficiency of 13.3% for nanowires with a diameter of 90 nm, 2.6 times than that without the grating structure.

The grating structure coupled to PDs can be not only a linear grating but also an angled grating. In 2015, Siampour et al. [59] reported a high-performance Si nanowires (SiNWs) PD operating at communication wavelength (Figure 7c). With a complex circular grating antenna structure, the light intensity in SiNWs was increased by five orders of magnitude. The simulation results showed that the device had a responsivity of 2.4 × 10^4^ A /W and a bandwidth of 3-dB over 300 GHz. Furthermore, the designed device structure could apply to other materials. In 2020, Luo et al. [60] reported an InP nanowires-based PD consisting of InP nanowires embedded in a dual-split bull’s eye (DSBE) plasmonic antenna (Figure 7d). The full-spectral responsivity and quantum efficiency of the device were enhanced owing to the presence of surface plasmons and their effective interaction with the guided modes in the NWs. The resultant PD exhibited a low noise equivalent power of 0.97 pW, a photoresponsivity of 0.96 AW^−1^ at 740 nm and an EQE of 163%.

### 2.3. Infrared Photodetectors with Nanoparticles

It has been proved that metal nanoparticles can be used to excite the surface plasmon resonance effect to improve PD performance. Specifically, metal nanoparticles, such as Au, Ag and Pt, possess a large number of free electrons. When the metal nanoparticles (NPs) are embedded in the light-absorbing material, the NPs coupled with the incident light to generate LSPR, then the light is trapped on the surface around the metal NPs [61,62]. Thus, the light absorption and photoelectric conversion efficiency of the neighboring material are enhanced.

#### 2.3.1. 2D Materials-Based Infrared Photodetectors with Nanoparticles

In 2014, Luo et al. [63] reported a highly-sensitive near-infrared (NIR) PD that was highly sensitive to 850 nm light illumination, and its on/off ratio obtained a breakthrough, reaching as high as 10^6^. In the structure of this device, multilayer graphene was located on the top of the SiNWs array, while the plasmonic AuNPs were evenly distributed on the graphene layers (Figure 8a). The addition of the AuNPs improved the efficiency of light capture between graphene sheets and Si NWs. Moreover, SPPs were also excited at the AuNPs/graphene interface and could effectively trap and guide light into the SiNWs. As evidenced by device analysis, the proposed PD achieved the responsitivity of ~1.5 AW^−1^ and detectivity of ~2.52 × 10^14^ cmHz^1/2^W^−1^ at zero bias voltage. Metal plasma nanostructures can also enhance the upconversion emission of upconversion nanoparticles (UCNPs) when placed in the vicinity of UCNPs. In the same year, Niu et al. [64] combined a monolayer 68 nm-diameter AuNPs with the graphene oxide (GO)-NaYF_4_:Yb, Tm@NaYF_4_ UCNPs nanocomposites they prepared. The AuNPs not only improved the absorption of graphene oxide layers but also increased the radiative emissivity of UCNPs and the absorption of Yb^3+^ ions. As a result, the photoresponsivity of the PD based on such materials to NIR light was greatly improved by about 10 times. The schematic illustration of this device is shown in Figure 8b. In 2015, Jang et al. [65] proposed a graphene-based PD, which achieved a photoresponsivity of 700A/W and photodetectivity of 10^13^ Jones in the wavelength range of 400–800 nm. In this device, the pentacene layer, deposited onto a graphene channel, functioned as a light-absorbing layer, while introduced AuNPs acted both as a charge-trapping layer and as a plasma light-scattering layer (Figure 8c). Under light illumination, the pentacene layer would produce electron-hole pairs and transfer light-induced hole carriers to the graphene layer. Next, the carriers in graphene were trapped by AuNPs, which also would store the incident light signal. The reported device enabled the further development of PDs with the function of photon signal storage.

Previously reported MoS_2_ phototransistors have low photoresponsivity due to poor optical absorbance. In 2015, Miao et al. [66] introduced Au plasmonic nanostructure on the top of few-layer MoS_2_ to result in the enhancement of photocurrent response in a MoS_2_ phototransistor. Due to the presence of Au nanoparticles, near-field oscillation enhancement and scattering effects appeared. The incident light was trapped in the Au nanostructures, leading to an enhanced local electric field. Thus, the light absorption of MoS_2_ was greatly increased. In their work, they fabricated MoS_2_ phototransistors with 4 nm thick AuNPs and periodic Au nanoarrays, respectively. As a result, the photocurrent response of the devices was improved by a factor of two and three, respectively. Coupling the 2D WS_2_ with the Au nanospheres, Liu et al. [67] built a plasmon resonance-enhanced WS_2_ PD in July 2019, which displayed good performance between the visible light and near-infrared region. Under this structure, the electric field on the WS_2_ films was strongly enhanced with the action of LSPR, contributing to more carriers being excited. The demonstrated PD exhibited good temperature stability after annealing at 300 °C. The structure of this detector is shown in Figure 8d. In the same month, Guo et al. [68] reported their efforts on a plasmon-enhanced MoS_2_ PD whose absorption in the wavelength range of 700–1600 nm was enhanced. The schematic illustration of the device is demonstrated in Figure 8e. Figure 8f shows the interface diagram of the device comprising AuNPs and MoS_2_ when operating in the infrared band. They deposited AuNPs on MoS_2_ layers by magnetron sputtering, which allowed them to control the physical form of AuNPs. The analysis showed that the photocurrent of this device was improved up to 480 nA, and the maximum photoresponsivity was 64 mAW^−1^ in comparison with a low current of 0.59 nA in MoS_2_-based PD without AuNPs. Besides, this PD had a fast response time and excellent stability.

In 2019, Rahmati et al. [69] designed a vertical-MoS_2_ nanostructure PD integrated with AuNPs (Figure 8g). In this study, they synthesized a colloid of AuNPs with an average diameter of 18.3 nm by laser ablation of Au target and decorated the top of MoS_2_ with them. It was revealed that the proposed PD had a 20-fold enhancement of the photocurrent over the dark current. In addition, a 2.1-fold increase in the device’s responsivity could be observed. In the same year, Jeon et al. [70] demonstrated a BP-based PD combined with AuNPs. These spherical AuNPs were embedded in amorphous silica shells with a 15nm silica cap and a 5nm Au core. During the device preparation process, they compared the effects of different annealing temperatures and durations on AuNP density and finally controlled the AuNP density between 6 × 10^7^ and 5 × 10^10^ cm^−2^ to make the LSPR maximized. With the assistance of LSPR, the BP/Au-NP PD had 60-fold and 500-fold improved photoresponsivity to visible and NIR wavelengths, respectively.

In 2021, Sun et al. [71] reported a plasmon-enhanced SnSe_2_ PD by introducing fused silica (SiO_2_) with embedded Ag NPs. The highlight of this structure was that the Ag NPs did not directly contact the SnSe_2_ layer, so the crystallinity of SnSe_2_ was not affected. The presence of Ag NPs contributed to the separation of electrons and holes. As a result, the performance of the proposed device was significantly enhanced. For example, an 881-fold improvement of the photoresponsivity was realized. In the same year, Nakazawa et al. [72] used Au nanorods with an aspect ratio of 5.9 to construct a photocurrent-enhanced PtSe_2_ PD whose plasmon peaks were in the NIR region. The Au nanorods were coated on 2D PtSe_2_ via the spin coating method. Besides, they also researched the effect of the nanorod solution’s concentration on the photoresponse of the proposed device. When the nanorod concentration was 5.0 nM, the photocurrent of the 2D PtSe_2_ /Au nanorod device was enhanced by five times that of the bare PtSe_2_ PD. A responsivity of 908 µW/A was observed at 0.5 V bias voltage.

Unlike others’ work, Li et al. [73] enhanced the optoelectric performance of a MoS_2_ PD utilizing the Au-MoS_2_-Au sandwich structure in 2022. The 60 nm AuNPs were obtained by annealing the samples with Au films deposited on the substrate at 450 °C for 30 min. Next, multilayer n-type MoS_2_ flakes were placed on the deposited AuNPs, and to this point, an Au-MoS_2_ PD was constructed (Figure 8h). The top AuNPs were made in the same way as the bottom AuNPs. Thus, the final Au-MoS_2_-Au PD was realized. It was found that the double-layer AuNPs had a stronger coupling effect than the single-layer Au NPs, which could contribute to a stronger surface electric field and higher light absorption, boosting the injection of more hot electrons into the conduction band of MoS_2_. Therefore, this PD had more excellent performance than the Au-MoS_2_ PD. For example, R_λ_ and D^*^ of the Au-MoS_2_-Au PD were increased to 1757 A/W and 3.44 × 10^10^ Jones, respectively.

**Figure 8 materials-16-03216-f008:**
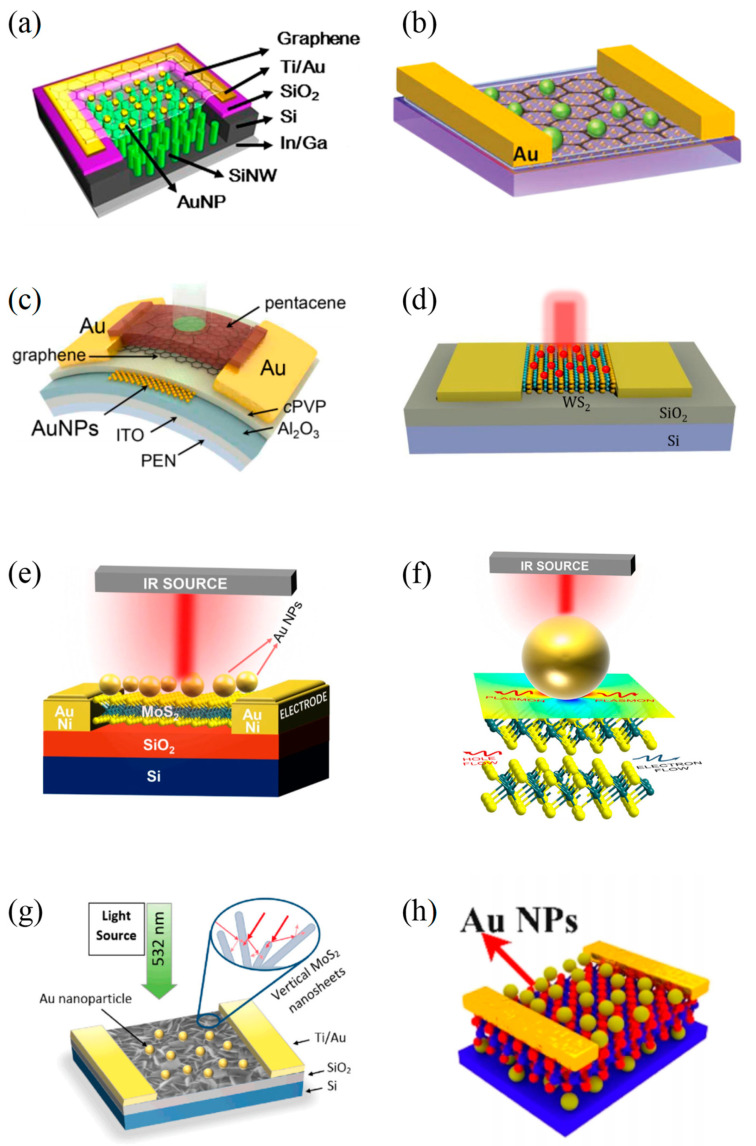
Structures of 2D materials-based infrared PDs with nanoparticles. (**a**) Schematic of the surface plasmon enhanced high-performance NIR PD. Copyright 2014, *Scientific Reports* [63]. (**b**) Schematic of the designed UCNPs and Au plasmon-enhanced NIR photoresponse device [64]. Copyright 2014, *Small*. (**c**) Schematic diagram of the device structure containing a graphene PD with a pentacene light absorption layer and an AuNP charge trapping layer. In the device, the ITO functions as a gate electrode and Au as source/drain electrode [65]. Copyright 2015, *Nano Letter.* (**d**) The 3D view of 1L-WS_2_-based PD. Au NPs (red nanospheres) solution was spun, coating the active area [67]. Copyright 2015, *Small*. (**e**) Schematic illustration of Au NPs covered MoS_2_ PD. The Au/Ni electrodes are used to apply the biasing voltage and collect photocurrent. (**f**) Schematic representation of interfaces composed of Au NPs and MoS_2_ in operation of the PD device at infrared wavelength. The electric field distribution is significantly different in various regions located at the interfaces of Au NPs and MoS_2_, resulting in a dramatic improvement in photo-responsibility [68]. Copyright 2019, *Applied Physics Letters*. (**g**) Schematic illustration of the proposed V-MoS_2_ PD [69]. Copyright 2019, *Applied Surface Science*. (**h**) Schematic diagram of the Au−MoS_2_−Au PD based on double-layered Au NP modification with sandwich structure [73]. Copyright 2022, *Applied Electronic Materials*.

#### 2.3.2. 1D Materials-Based Infrared Photodetectors with Nanoparticles

In 2013, Zhou et al. [74] demonstrated a plasmonic enhanced carbon nanotube (CNT)-based PD with AuNPs, whose photocurrent was increased by three times. In their work, they coupled the dodecanethiol-capped AuNPs obtained by the self-assembly method with the CNT. From the extinction spectrum, it could be seen that there was a plasmon resonance peak at about 631 nm, and the peak absorption rate was 48%. They also researched the effect of CNT diameter on the local plasma enhancement of AuNPs and found that the larger the diameter of CNT, the stronger the field enhancement effect. The principle of field enhancement on CNT owing to Au nanoparticles is demonstrated in Figure 9a. Since Si has an indirect band gap, it has poor light absorption properties in the NIR range. In 2014, Jee et al. [75] proposed a solution to this problem using plasmonic hemispherical AuNPs in a SiNWs PD. Owing to the role of the surface plasmon resonance effects and the Schottky contact formation between SiNWs and AuNPs, this device exhibited strong NIR light absorption and high responsivity. The responsivity spectra for SiNWs with and without AuNPs are shown in Figure 9b.

In 2022, Qi et al. [76] verified that introducing AuNPs in a single-walled CNT-Ge NIR PD could improve the photoelectric performance. Due to the presence of the AuNPs, the CNT films’ work function and surface plasmon resonance were improved, resulting in a higher photoresponse in the wavelength range of 1250–2000 nm (Figure 9c). The experimental results revealed that the proposed PD possessed a responsivity of 476 mAW^−1^ and a detectivity of 1.0 × 10^12^ cm Hz^1/2^W^−1^. In the same year, Zhu et al. [77] placed capsule-like AuNPs with an absorption edge of 800 nm onto the GaAs NWs PD (Figure 9d). Thanks to the AuNPs, the absorption intensity of GaAs NWs was improved, especially near 800 nm. As a result, the performance of the treated PD had a good improvement. For example, the responsivity and detectivity were up to 6.56 A W^−1^ and 5.6 × 10^10^ Jones, improved by 3.2 and 2.53 times, respectively.

#### 2.3.3. 0D Materials-Based Infrared Photodetectors with Nanoparticles

Although PbS QDs PDs can achieve broad-spectrum detection from UV to NIR, the response rate in the NIR band is relatively low. In 2012, Arquer et al. [78] demonstrated that the embedding of Ag metal nanoparticles (MNPs) could result in a responsivity increase in PbS colloidal QDs (CQDs) PDs. The hemispherical Ag MNPs they prepared had an average diameter of 100 nm and a height of 30 nm and were distributed discretely on the glass substrates with a surface coverage of 31%. The results found that in the NIR range, there was a maximum 2.4-fold increase in responsivity near the edge of the absorption band of 1 μm for a 400 nm thick PD. The team also used a glass slide coated with 200 nm of Ag as a reflector to enhance the reflectance in the 400–1200 nm region, and the responsivity was consequently further improved. The structural diagram of this device is shown in Figure 10a. Few of the previously reported methods can simultaneously meet the requirements of increasing photocurrent and decreasing dark current of PbS CQD PDs. Hence, in 2014, He et al. [79] mixed 0.5% and 1% by weight of Ag nanocrystals (AgNCs) with PbS CQDs in solution and then constructed PbS CQD/AgNCs NIR photoconductive PDs by layer-by-layer spin coating method (Figure 10b). Since AgNCs were able to capture the photogenerated electrons in the conduction band of PbS CQDs and extend carrier lifetime, the photocurrent was significantly increased. With the synergistic effects of PbS CQDs and AgNCs, the device had a slightly reduced dark current and a ~2.4-fold improvement in detectivity. The team further made a flexible and sensitive PD based on the PbS CQD/AgNCs composite nanomaterials, which provided a novel strategy for photodetection applications requiring lightweight and mechanical flexibility. The performance of the flexible detector under different bending angles is shown in Figure 10c.

In 2014, Chen et al. [80] prepared 40–50 nm wide and 90–110 nm long Au nanorods (Au NRs) with the seed-mediated growth procedure and embedded them into HgTe QDs/ZnO heterojunction PDs (Figure 10d). They investigated the effect of the position of Au nanorods relative to the photoactive layer on the absorption and charge generation/transport processes of HgTe QDs/ZnO PDs in the NIR wavelength range. In this structure, the two main elements that absorbed the incident light were the HgTe QDs layer and the Au nanorods. There were resonant collective electron oscillations within the Au nanorods, making the local electromagnetic field enhanced so that the absorption of HgTe QDs was increased. This behavior made the absorption of the HgTe QDs layer vary with the thickness of the ZnO top coating. Furthermore, the nanorods also had a strong absorption capacity. In 2019, Zhao et al. [81] exploited the Au-PbS core-shell nanorods (Au-PbS-NRs) synthesized by solution growth of PbS shells onto Au nanorod cores to fabricate a plasmon-enhanced NIR PD (Figure 10e). Because the LSPR of Au NRs enhanced the local electromagnetic field, electrons in the valence band of PbS were excited to the conductive band when the incoming light hit, which provided more convenience for the excited electrons in the PbS shell into the Au core. The experimental results showed that under an 808 nm laser light, the responsivity, detectivity and EQE of this device reached 18.5 A W^−1^, 1.22 × 10^11^ Jones and 2844%, respectively. In the next year, Zhao et al. [82] reported another sandwiched PbS/Au/PbS NIR phototransistor utilizing PbS CQDs and Au NRs as active materials (Figure 10f). With the action of the LSPR of Au NRs, the device had a significantly higher photocurrent, 2.6 times than that of the bare PbS QDs PD. The EQE of the PD was 2.75 times higher than that of the PbS QDs PD at 1020 nm wavelength. In general, this device achieved a photoresponsivity of 8.2 A W^−1^, detectivity of 6.3 × 10^10^ Jones and EQE of 1251% with the 808 nm laser illumination. The design of this structure provided a good scheme for enhancing the response of PbS QDs PDs in the NIR region.

### 2.4. Infrared Photodetectors with Metal Array Structures

In metal array structures, near-field local effects of the plasmonic nanostructure increase the optical field density near the nanostructure on the one hand, and the coupling effect between the plasmonic metal nanomaterials increases the optical field density in the coupling region on the other hand. Based on this principle, plenty of infrared PDs with metal array structures have been fabricated in recent years.

#### 2.4.1. 2D Materials-Based Infrared Photodetectors with Metal Array Structures

To intensify the photoelectric conversion efficiency of the graphene-based PDs, in June 2012, Fang et al. [83] designed a graphene-antenna sandwich PD (Figure 11a). This device could be effectively used for photoelectric detection in the visible and near-infrared regions, where it achieved an internal quantum efficiency of nearly 20% and a photocurrent enhancement effect of 800%. In addition to the photoelectrons excited by the graphene layer itself under the action of the locally enhanced electric field, the hot electrons generated by the surface plasmon excitations during the decay process also promoted the increase of the detector’s photocurrent. Based on this work, Fang et al. also proved that the plasmon could be used for graphene doping in September 2012. This doping was accomplished by hot electrons generated by the plasmonic antenna arrays. Moreover, the degree of this doping could be regulated by changing the size of the plasmon antenna, the incident laser wavelength and the laser power density. This photo-induced doped graphene novel hybrid material they discovered had broad application prospects in photoelectric devices [84].

As Fano resonance can enhance the absorption and promote the generation of the hot electron, it is suited to increase the photocurrent of plasmonic enhanced PDs. In February 2020, Zhang et al. [85] placed the monolayer WS_2_ and monolayer WSe_2_ in sequence on the top of the periodic Au nanorod arrays (Figure 11b). In their work, they researched the Fano resonances from plasmon-exciton coupling in this structure, the asymmetry of which could be controlled by adjusting the polarization of the incident light. The Fano resonance of this heterogeneous structure contributed to the development of novel micro and nano optoelectronic devices. Traditional direct deposition procedures of Au nanoarrays can destroy the surface crystal lattice of semiconductor materials, which hinders the enhancement of PDs’ responsivity. In December 2020, using the integration strategy of physical transfer, Yang et al. [86] reported a surface plasmonic enhanced BP PD by introducing the Au squares’ nanoarray (Figure 11c). As a result, a remarkable 1000% improvement in photoresponse was achieved. What’s more, this physical transfer realized the damage-free metal-semiconductor interface.

In 2020, Gu et al. [87] proposed a hybrid PD based on perovskite-metallic nanostructures, where Au arrays were introduced (Figure 11d). The hybridization with Au arrays could effectively limit the recombination of photo-induced electron-hole pairs, which was due to the increase of electron transfer between the film and Au array and the electron transfer pathway on the metal surface. Moreover, the Au arrays could also promote the generation of photoelectrons and holes thanks to the enhanced field of local surface plasmon resonance. As a result, the responsivity and EQE of the detector were 51mA/W and 12%, respectively. However, the maximum optical response enhancement value of ordered structure devices was about 4–5 fold, which was originally lower than that of disordered structures. Therefore, more research on the ordered plasmonic structure needs to be completed. Through theoretical analysis and experimental comparison of two kinds of plasma structures, namely nano-orthorhombic and nano-disc, in 2021, Guskov et al. [88] improved the efficiency of 2D semiconductor-based photodetection with ordered plasmonic excitonic structures. Under the optimal conditions, the photocurrent enhancement and the maximum increase of photosensitivity reached almost ten times. The maximum photosensitivity of the prepared device was about 0.1 A W^−1^.

#### 2.4.2. 1D Materials-Based Infrared Photodetectors with Metal Array Structures

In 2014, Casadei et al. [89] demonstrated that coupling the Au dimer nanoantenna into individual GaAs NWs could produce photon-plasma resonance. In addition, it was also proved that the nonlinear optical responsivity of single GaAs NWs could be tuned by designing the near-field coupling mechanism, which was conducive to the development of high-resolution detectors. Utilizing the glancing angle deposition (GLAD) technique, in 2021, Yadav et al. [90] fabricated the 200 nm TiO_2_ NW array and 5 nm In NPs on the top of 70 nm TiO_2_ thin film, and at the top were Ag electrodes, all of which were performed on the *p*-type Si substrate. Their work proved that In NPs assumed significant jobs in the enhancement of the PDs’ photosensing despite a short decay time of 0.56 s. Compared with the reference device without In NPs, the designed device with In NPs exhibited maximum current conduction of 10.81 μA/cm^2^, increased by 1.3 times, which benefited from oxygen-related trap states at the edge of Ag Schottky contact. And it also could be observed that the photosensitivity was enhanced by an average of ~3.6 fold under the reverse bias.

#### 2.4.3. 0D Materials-Based Infrared Photodetectors with Metal Array Structures

In 2016, Tang et al. [91] reported a plasmon resonance enhanced HgSe CQDs PD by integrating HgSe CQDs films with Au nanodisk arrays (Figure 12a). The Au nanodisk arrays coupled the incoming infrared light to a collective photon mode in one plane, resulting in strong and narrow spectral resonance features. Tuning the resonant wavelength to the central wavelength of the narrowband PD by controlling the radius of the disks and the distance between the disks could greatly enhance the optical response at the central wavelength. In this work, they made four narrowband PDs whose center wavelengths were 4.2 μm, 6.4 μm, 7.2 μm and 9.0 μm, respectively. With the assistance of Au disks, the responsivity of central wavelength was increased by 517% at 4.2 μm, 288% at 6.4 μm, 257% at 7.2 μm and 208% at 9.0 μm to 145 mA W^−1^, 92.3 mA W^−1^, 88.6 mA W^−1^ and 86 mA W^−1^. Likewise, in 2018, Tang et al. [92] also added plasmon nanodisk arrays in a HgTe CQDs MWIR photovoltaic detector to improve the optical collection efficiency of HgTe CQDs absorption films (Figure 12b). Besides, an interference structure on top was used to optimize the device since the reflected light played an essential part in an interference cavity. As a result, the responsivity and the detectivity of the designed PD were significantly improved. For instance, at 4.5 μm, the low-temperature responsivity was 1.62 A/W, which was a 2–3 times improvement (Figure 12c). And the detectivity reached 10^10^ Jones at 4 μm and 220 K.

Yakimov et al. [93] demonstrated that Al nanodisks could enhance the NIR optical response of the Ge/Si QDs PD via LSPR in 2020 (Figure 12d). The proposed Ge/Si QDs device was grown on the silicon-on-insulator (SOI) substrate with an active region that included 5-layer Ge QDs, but each layer was separated by a 10 nm Si barrier. The 2D Al disk array, whose lattice constant was 400 nm and showed square lattice symmetry, was put on top of multi-layer Ge/Si QD heterostructures. This device had the dual advantages of a metal-disk-excited local surface plasma and radiative coupling with the SOI waveguide mode. The team compared the bare QD PD with the QDs PDs that had periodic Al disk arrays with different disk diameters, such as 150 nm, 175 nm, 200 nm and 225 nm. As a result, they noticed that the appropriate selection of the period of the Al disk array and the disk diameter contributed to the detection efficiency, which was improved by 40 times at 1.2 μm and by 15 times at 1.55 μm.

In conclusion, based on different plasma structures, such as hole arrays, gratings, nanoparticles and metal arrays, plasmonic enhanced nanocrystal infrared PDs have been studied extensively in the past several decades. Meanwhile, benefiting from the excellent characteristics of low-dimensional materials, infrared PDs exhibit better performance. In the future, improving the performance of PDs through the plasma exciton effect is an important way to the development of PDs. The performance of PDs in the references discussed above is given in Table 1. In addition, we further summarize the peak-specific detectivities of some representative plasma-enhanced infrared PDs, as shown in Figure 13.

## 3. Conclusions and Outlook

The surface plasmon resonance effect could enhance the light absorption of materials. In this work, by summarizing the plasmonic enhanced nanocrystal infrared PDs, we show that the performance of PDs has been significantly improved due to the addition of plasmonic structures. For example, the responsivity and quantum efficiency of detectors can be effectively improved, and the footprint of the device can be reduced. Moreover, such PDs have effectively advanced the progress of PDs operating at high temperatures.

With decades of development, plasma exciton technology has become more mature. However, traditional plasmon materials are generally precious metals, such as Au and Ag, which have certain defects. First of all, from the perspective of cost, precious metals are expensive and, thus, not suitable for the development of large-scale production devices. Next, precious metals have a low melting point, so nanostructures made of these metals are easily deformed at high temperatures. Hence, it is necessary to search for novel materials for plasmon applications. For example, Ge and Al are promising candidates. Moreover, micro-nano structures with different shapes and sizes have different effects on the excitation and application of plasmons on the device surface. With the continuous development of manufacturing technology, nanostructures with more complex 3D geometry can be realized. This will further increase the light limit in the absorber, broaden the response band and further improve the detection performance.

## Figures and Tables

**Figure 1 materials-16-03216-f001:**
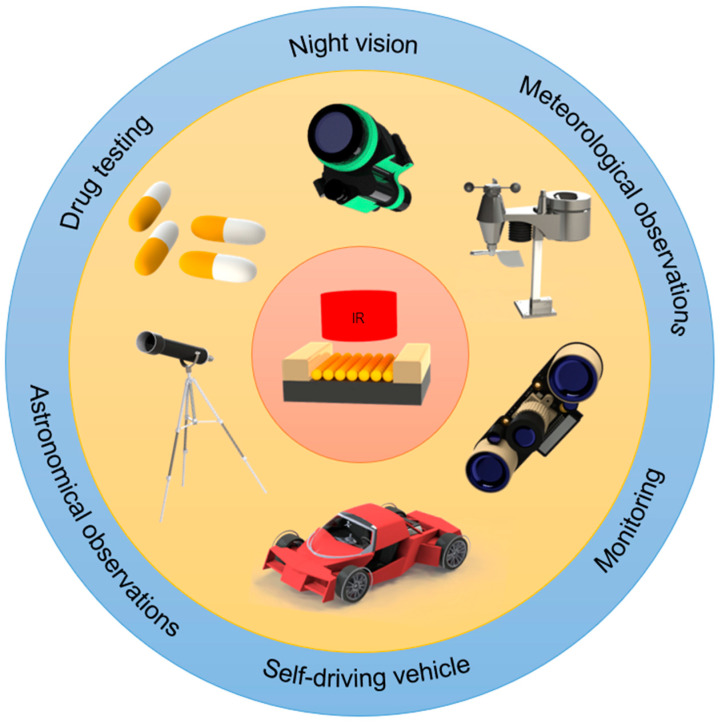
Potential applications of infrared PDs.

**Figure 2 materials-16-03216-f002:**
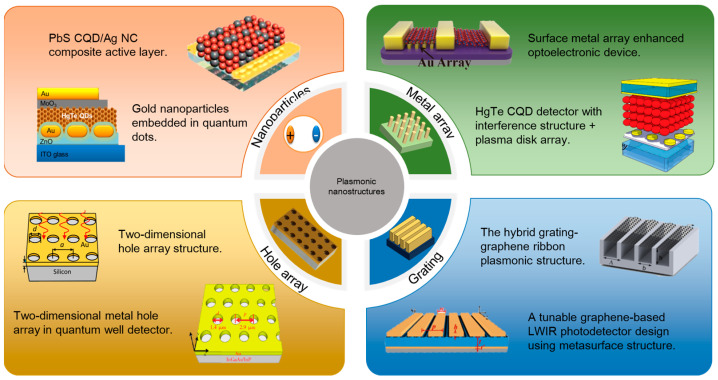
Progress in plasmonic enhanced infrared PDs. These four parts represent infrared PDs based on different plasmon-enhanced structures, namely nanoparticles, metal grating, hole array and metal array.

**Figure 3 materials-16-03216-f003:**
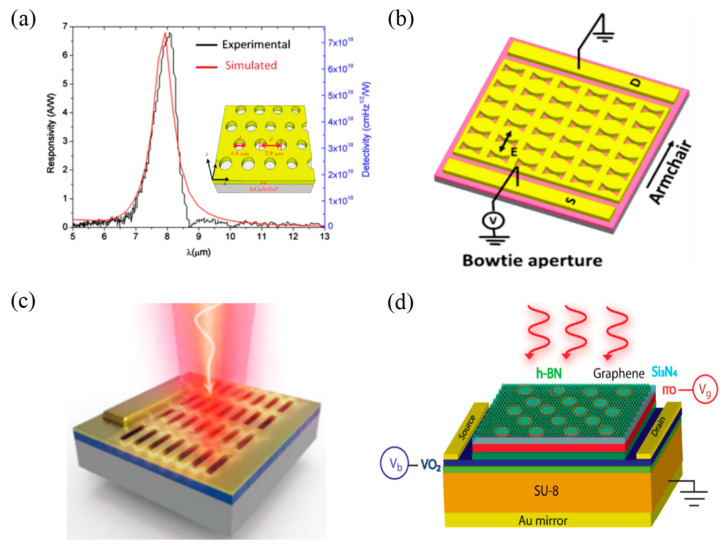
Structures and performance characterization of 2D materials-based infrared PDs with hole array structures. (**a**) Schematic diagram of the device’s responsivity and detectivity. The inset shows a schematic of the structure of the device with an array of metal vias [34]. Copyright 2010, *Applied Physics Letters*. (**b**) Schematic diagram of the BP device with the bowtie hole array [35]. Copyright 2018, *ACS Nano.* (**c**) Polarization structure of the nanohole array [36]. Copyright 2020, *Nanoscale*. (**d**) Schematic diagram of the proposed mid-infrared PD based on IMT [37]. Copyright 2021, *ACS Applied Nano Materials*.

**Figure 4 materials-16-03216-f004:**
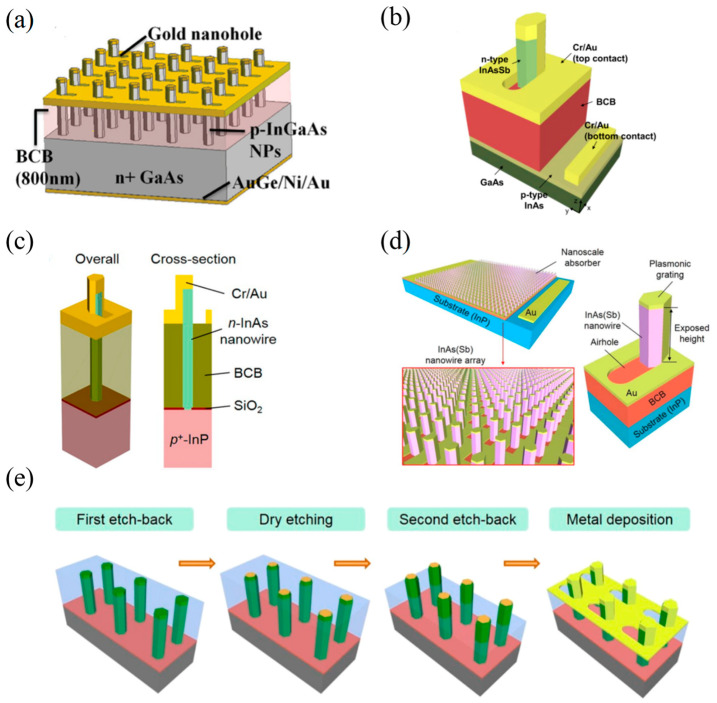
Structures and performance characterization of 1D materials-based infrared PDs with hole array structures. (**a**) Schematic of the NP PD array [38]. Copyright 2011, *Nano Letters.* (**b**) Schematic of the proposed NP PD structure [39]. Copyright 2015, *Nano Letters.* (**c**) Schematics of the unit cell of an InAs nanowire PD [40]. Copyright 2018, *Nano Letters.* (**d**) Schematic diagrams of the nanowire-based InAs(Sb)-InP heterojunction photodiode with the InAs(Sb) nanowire array as a photoabsorber [41]. Copyright 2018, *Nanotechnology*. (**e**) Process for the preparation of nanowires with Al_2_O_3_ passivation shells. Four major steps are shown [42]. Copyright 2019, *Nano Letters*.

**Figure 5 materials-16-03216-f005:**
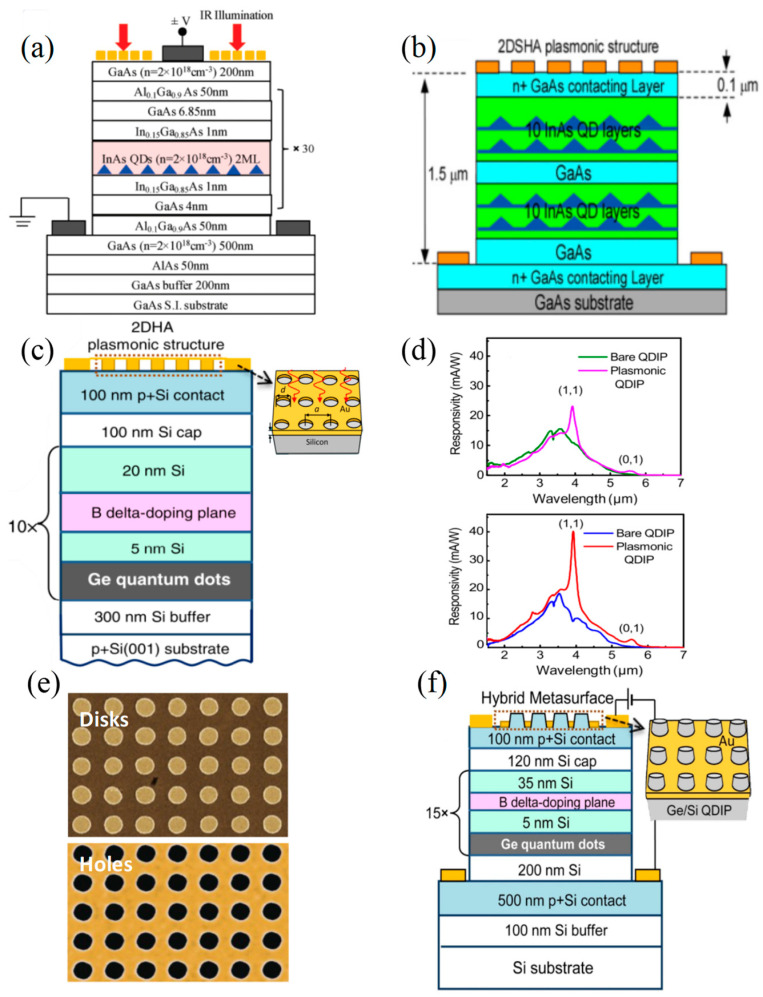
Structures and performance characterization of 0D materials-based infrared PDs with hole array structures. (**a**) Schematic diagram of the structure of the 2DHA infrared detector [44]. Copyright 2011, *Nano Letters.* (**b**) Schematic diagram of the 20-QDIP with a top surface plasma structure [45]. Copyright 2012, *Journal of Physics D: Applied Physics*. (**c**) Schematic diagram of the top 2DHA plasma structure enhanced with QDIP [46]. Copyright 2017, *Journal of Applied Physics.* (**d**) Comparison of the spectral response of QDIP with 2DHA plasmonic structure and bare QDIP under substrate side and top irradiation [47]. Copyright 2017, *Optics Express*. (**e**) Scanning electron microscopy image of a round Au disk and hole in an Au membrane [48]. Copyright 2018, *Optical Materials Express.* (**f**) Schematic diagram of the QDIP structure enhanced with the top hybrid metasurface [49]. Copyright 2020, *Journal of Physics D: Applied Physics*.

**Figure 6 materials-16-03216-f006:**
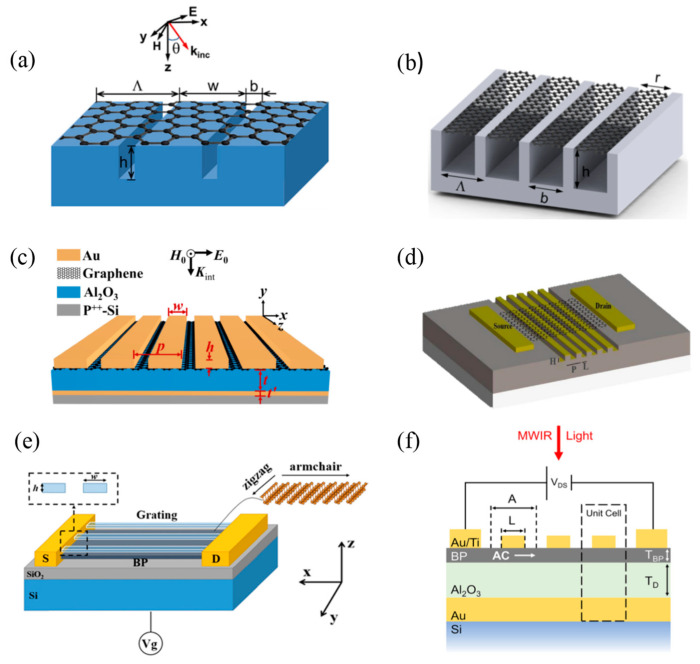
Structures of 1D materials-based infrared PDs with grating structures. (**a**) Schematic of the graphene-covered deep Ag grating [51]. Copyright 2014, *Applied Physics Letters*. (**b**) Schematic of the hybrid grating-graphene ribbon plasmonic structure under plane wave incidence [52]. Copyright 2015, *ACS Photonics.* (**c**) Schematic structure of the graphene-based PD [53]. Copyright 2018, *Optics Express*. (**d**) Schematic diagram of the structure of a metamaterial integrated graphene PD device. The metamaterial is designed as a grating structure [54]. Copyright 2019, *Applied Surface Science.* (**e**) Schematic diagram of 20-layer BP-based PD [55]. Copyright 2019, *Optical Materials Express*. (**f**) Schematic diagram of the proposed device with a raster structure and only three individual unit cells [56]. Copyright 2022, *Nano Letters*.

**Figure 7 materials-16-03216-f007:**
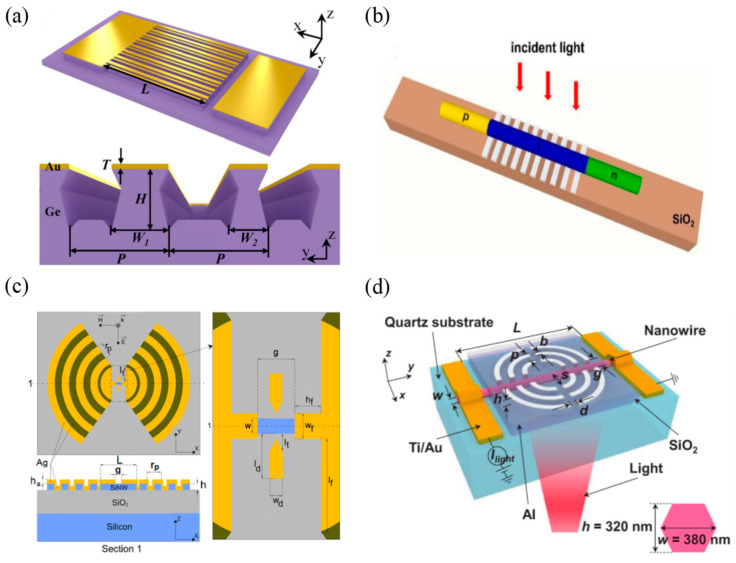
Structures and performance characterization of 1D materials-based infrared PDs with grating structures. (**a**) Schematic illustration of the dual-band plasmon-enhanced PD with alternate Au grating-Ge NW arrays [57]. Copyright 2021, *Applied Physics Letters*. (**b**) Schematic diagram of the laterally oriented NW solar cells with Ag gratings [58]. Copyright 2021, *Nanomaterials*. (**c**) Schematic diagram of the designed plasma antenna structure [59]. (**d**) Schematic of the InP NW PD fabricated with DSBE antenna [60]. Copyright 2020, *Advanced Optical Materials*.

**Figure 9 materials-16-03216-f009:**
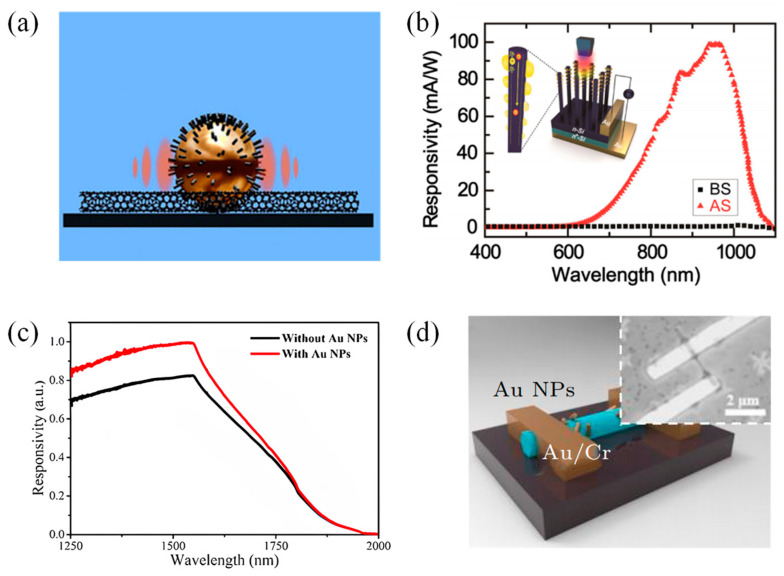
Structures and performance characterization of 1D materials-based infrared PDs with nanoparticles. (**a**) Plasmon coupling between Au nanoparticle and CNT when the polarization of the incident light is parallel to the CNT axis [74]. Copyright 2013, *Applied Physics Letters*. (**b**) Responsivity spectra for bare SiNW (black) and Au NP-coated SiNW (red) PDs. (Inset is a schematic illustration of the proposed PD consisting of Au hNP-coated SiNWs.) [75]. Copyright 2022, Nano Convergence. (**c**) Spectral response of the SWCNT/Ge PD with and without AuNPs [76]. Copyright 2013, *Photonics* (**d**) Schematic of a single GaAs NW PD; the inset is the SEM image [77]. Copyright 2022, *Chinese Physics B*.

**Figure 10 materials-16-03216-f010:**
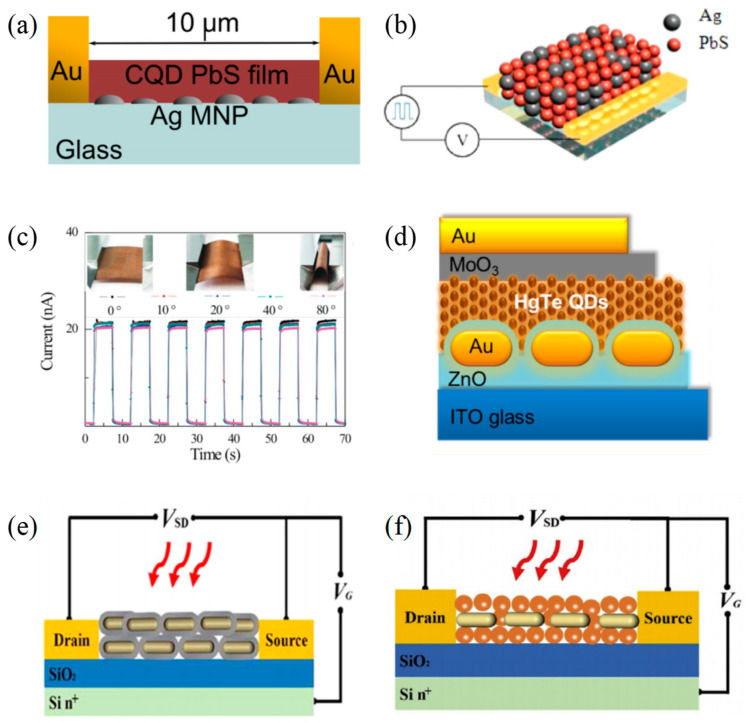
Structures and performance characterization of 0D materials-based infrared PDs with nanoparticles. (**a**) Schematic of the PD device structure with embedded nanoparticles [78]. Copyright 2012, *Applied Physics Letters*. (**b**) Schematic demonstration of device configuration. Red spheres represent PbS CQDs, and black spheres represent Ag NCs. (**c**) Paper-based PDs through many cycles of bending and unbending at wide angles (20–80°). The inset shows device photographs of the bending state [79]. Copyright 2014, *ACS Photonics*. (**d**) Schematic diagram of the proposed HgTe QDs photodiode based on plasmonic Au nanorod structures [80]. Copyright 2014, *ACS Nano.* (**e**) Schematic diagram of the field-effect phototransistor [81]. Copyright 2019, *Journal of Materials Science*. (**f**) Schematic diagram of the designed Au NRs-PbS QDs phototransistor [82]. Copyright 2022, *Journal of Alloys and Compounds*.

**Figure 11 materials-16-03216-f011:**
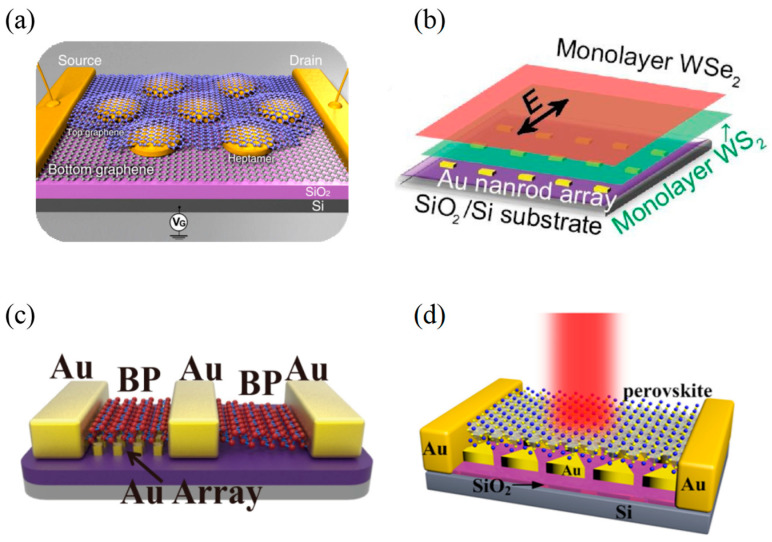
Structures and performance characterization of 2D materials-based infrared PDs with nanoarray structures. (**a**) Schematic diagram of the proposed graphene–antenna sandwich PD [83]. Copyright 2012, *Nano Letter*. (**b**) Schematic representation of WS_2_ and WSe_2_ heterogeneous bilayers on periodic Au nanorod arrays [85]. Copyright 2020, *Photonics Nanostructure*. (**c**) Schematic diagram of the BP PD with metal array structure [86]. Copyright 2021, *IEEE Trans Electron Devices*. (**d**) Schematic diagram of the hybrid PD combined with the Au triangle array [87]. Copyright 2021, *Materials & Design*.

**Figure 12 materials-16-03216-f012:**
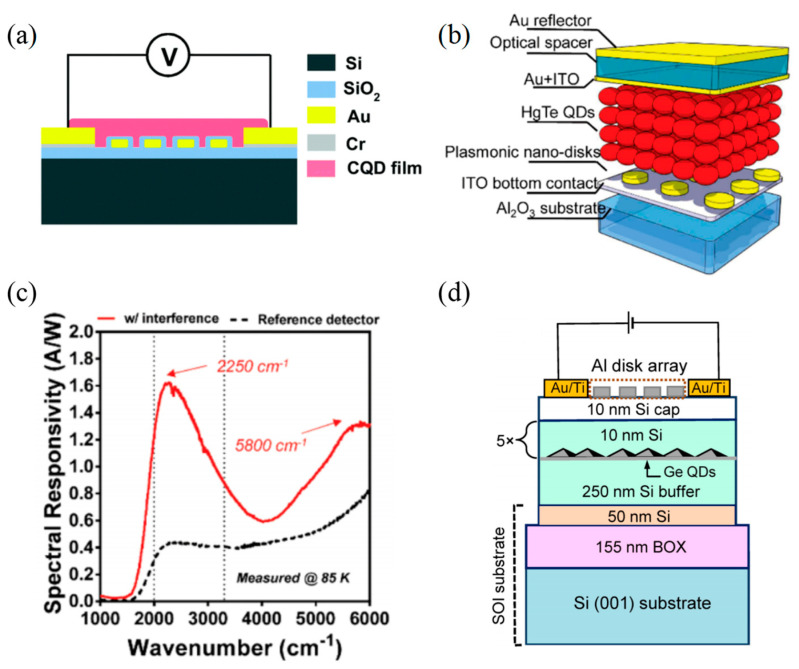
Structures and performance characterization of 0D materials−based infrared PDs with nanoarray structures. (**a**) Schematic configuration of HgSe QDs−based infrared PD with plasmonic Au structures [91]. Copyright 2017, *J Mater Chem C Mater*. (**b**) Schematic illustration of the HgTe CQD detector with interference structure + plasmonic disk array. (**c**) Measured spectral responsivity of HgTe CQDs with interference-enhanced plasmonic disk array [92]. Copyright 2018, *ACS Nano*. (**d**) Layer sequence of the five-period Ge/Si-on-SiO_2_ lateral QDPs [93]. Copyright 2020, *Journal of Applied Physics*.

**Figure 13 materials-16-03216-f013:**
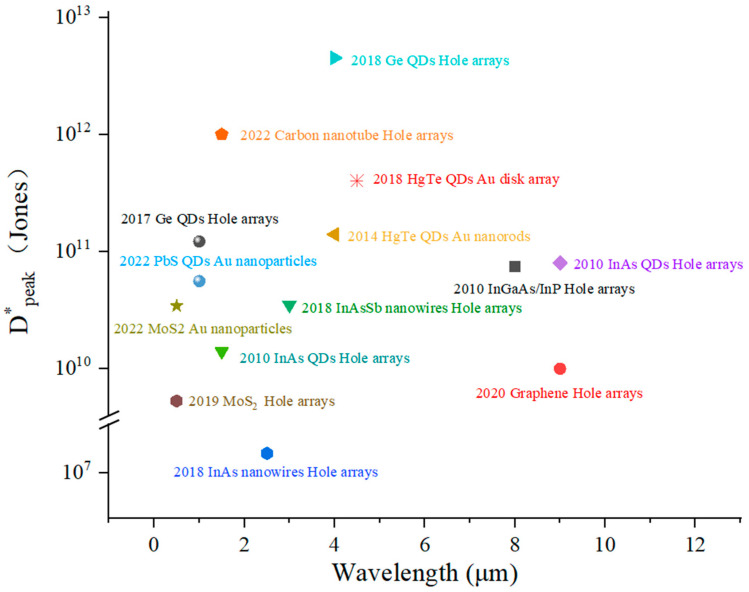
Peak specific detectivities of some representative plasma-enhanced infrared PDs.

**Table 1 materials-16-03216-t001:** Progress in plasmonic enhanced nanocrystal infrared PDs.

Year	Plasma Structure	Nanomaterials Used	Spectral Range (μm)	Responsivity (A/W)	$D^{*}$ (Jones)	EQE/ IQE (%)	Ref.
2010	Hole arrays	InGaAs/InP	<8	7	7.4 × 10^10^	-	[34]
2020	Hole arrays	graphene	<12	-	~10^10^	-	[36]
2011	Hole arrays	InGaAs nanopillars	NIR	0.28	-	-	[37]
2016	Hole arrays	InAsSb nanowires	<3	0.194	-	29%	[38]
2018	Hole arrays	InAs nanowires	SWIR	~1.5 × 10^−3^	2.5 × 10^7^	-	[39]
2018	Hole arrays	InAs(Sb) nanowires	SWIR-MWIR	0.75	3.5 × 10^10^	-	[40]
2010	Hole arrays	InAs QDs	<12	1.02	8 × 10^10^	-	[44]
2017	Hole arrays	Ge QDs	<6	0.04	1.4 × 10^11^	-	[47]
2018	Hole arrays	Ge QDs	<6	0.42	4.5 × 10^12^	2%	[48]
2022	Au grating	Black phosphorus	MWIR	0.077	-	4%	[56]
2021	Au grating	Ge nanowire	NIR	0.75	-	-	[57]
2020	Circular grating	InP Nanowire	NIR	0.96	-	163%	[60]
2014	Au nanoparticles	Graphene	NIR	1.5	2.52 × 10^14^	-	[63]
2015	Au nanoparticles	Graphene	0.4–0.8	700	10^13^	-	[65]
2019	Au nanospheres	WS_2_	Visible-NIR	1050	-	-	[67]
2019	Au nanoparticles	MoS_2_	0.7–1.6	0.064	-	-	[68]
2019	Au nanoparticles	MoS_2_	-	0.024	5.3 × 10^9^	-	[69]
2022	Au nanoparticles	MoS_2_	-	1757	3.44 × 10^10^	4106%	[73]
2014	Au nanoantennas	Si nanowire	NIR	0.1	-	-	[75]
2022	Au nanoparticles	Carbon nanotube	SWIR	0.476	1 × 10^12^	-	[76]
2022	Au nanoparticles	GaAs nanowire	NIR	6.56	5.6 × 10^10^	-	[77]
2012	Ag nanoparticles	PbS CQDs	NIR	~300	-	-	[78]
2014	Ag nanocrystals	PbS CQDs	NIR	0.0038	7.1 × 10^10^	~11%	[79]
2014	Au nanorods	HgTe CQDs	NIR-MWIR	-	~1.5 × 10^10^	~7%	[80]
2019	Au nanorods	PbS CQDs	NIR	18.5	1.22 × 10^11^	2844%	[81]
2020	Au nanorods	PbS CQDs	NIR	8.2	6.3 × 10^10^	1251%	[82]
2012	nanoantenna	Graphene	NIR	0.013	-	22%	[83]
2020	Au triangle arrays	perovskite	Visible-NIR	0.051	-	12.6%	[87]
2016	Au disk arrays	HgSe CQDs	MWIR	0.1487	-	-	[91]
2018	Au disk arrays	HgTe CQDs	MWIR	1.62	4 × 10^11^	45%	[92]
2020	Al disk arrays	Ge/Si QDs	NIR	0.05	-	-	[93]

## Data Availability

Not applicable.

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
