# Peer review of "Plasmonic Enhanced Nanocrystal Infrared Photodetectors"

_materials, 2023, doi:10.3390/ma16083216_

Round 1
Reviewer 1 Report
The manuscript discusses the well-known plasmonic effects and their application in improving the performance of infrared photo detectors. In the manuscript, for the plasmonic effect, typical noble metals like gold and silver are considered, and also newly used materials like Germanium and Aluminium are also discussed. The effect of various structures on 0d, 1d, 2D and 3D materials is also discussed.
Some of the things that can be improved
1. Include other 2D materials like SnSe, Sb2Se3 etc also. You have discussed Us2 etc
2. Mostly plasmonic effects are good in collecting light in visible region. Why it is effective in the IR range is not explained anywhere. Thus, the mechanism part is missing
3. Please could you explain the manuscript through your own image and diagram? Summarize the result with some creative diagram; then it can be a good review. Right now, the manuscript seems t be simply a collection of data
Mostly it is OK.
Author Response
Please refer to the attachment for specific reply.

Reviewer 2 Report
- Can you summarize the potential applications of Therefore, infrared photodetectors in a figure?
- -line 48, “Plasmonic enhanced photodetectors can enhance the local electric field intensity 54 which is conducive to the effective separation of electrons and holes,” I think here you can mention the hot coupling effect for antennas applications
[https://doi.org/10.1016/j.ijleo.2018.07.135],
X. Li et al., Hybrid nanostructures of metal/two-dimensional nanomaterials for plasmon-enhanced applications Chem. Soc. Rev.(2016)
- Many typing errors should be reconsidered
- Line 182, what is 40?
- Once you used abbreviations, no need to repeat them. For example; quantum dots line 193, 28
- In lie 204, “Error! Reference source not found”, please check it
- Line 213, what is 43? Please check carefully the references
- Line 213,” 6 times photo- 212 current enhancement”. Using values is always better for a good comparison
- Photodetectors were repeated everywhere, you can use PD
- Line 236, same ref problem 47
- Fig 4, (e), the actual reasons for using disks, and the hole may be explained
- Line 258, “based on the deep metal (Ag)” is not correct, metal is not Ag, you have to be more defied
- Line 267, “long-wave 266 infrared (LWIR)”, was repeated before as well
- Line 276, ref 52?? not the format of the journal, 309
- Line 321, 322, no need to take copyrights for open access journal even is also MDPI
- Line 355 “Error! Refer- 355 reference source not found..”364, 373,
- “2.3.1 2D Materials-based Infrared Photodetectors with Nanoparticle Structure” do you man doping?
- Error! Reference source not found.. still appears so much in the whole manuscript
- Line 409, 411, sometimes the name of the figure is between () and others not
- Line 419, no need for the month
- Line 446, “2.3.2 1D Materials-based Infrared Photodetectors with Nanoparticle Structures” the meaning is not clear, the nanoparticles are always the same, or nanostructures
- Lin 683, in the table “2844%” can you check this value
I think the English must be revised well, many typing and Grammarly notice there
Author Response

(The authors gave the same response as above.)

Round 2
Reviewer 1 Report
can be accepted
moderate editing required
Reviewer 2 Report
accept
acceptable